# Longitudinal Humoral Responses after COVID-19 Vaccination in Peritoneal and Hemodialysis Patients over Twelve Weeks

**DOI:** 10.3390/vaccines9101130

**Published:** 2021-10-04

**Authors:** Claudius Speer, Matthias Schaier, Christian Nusshag, Maximilian Töllner, Mirabel Buylaert, Florian Kälble, Paula Reichel, Julia Grenz, Caner Süsal, Martin Zeier, Paul Schnitzler, Christian Morath, Katrin Klein, Louise Benning

**Affiliations:** 1Department of Nephrology, University Hospital Heidelberg, 69120 Heidelberg, Germany; matthias.schaier@med.uni-heidelberg.de (M.S.); christian.nusshag@med.uni-heidelberg.de (C.N.); maximilianconstantin.toellner@med.uni-heidelberg.de (M.T.); mirabel.buylaert@med.uni-heidelberg.de (M.B.); florian.kaelble@med.uni-heidelberg.de (F.K.); paulareichel@web.de (P.R.); julia.grenz@med.uni-heidelberg.de (J.G.); Martin.Zeier@med.uni-heidelberg.de (M.Z.); christian.morath@med.uni-heidelberg.de (C.M.); katrin.klein@med.uni-heidelberg.de (K.K.); 2Molecular Medicine Partnership Unit Heidelberg, EMBL, 69120 Heidelberg, Germany; 3Institute of Immunology, University Hospital Heidelberg, 69120 Heidelberg, Germany; caner.suesal@med.uni-heidelberg.de; 4Department of Infectious Diseases, Virology, University Hospital Heidelberg, 69120 Heidelberg, Germany; paul.schnitzler@med.uni-heidelberg.de

**Keywords:** COVID-19, COVID-19 vaccination, hemodialysis, peritoneal dialysis, humoral response, BNT162b2, vaccination strategy, SARS-CoV-2

## Abstract

It has been demonstrated that patients on hemo- or peritoneal dialysis are particularly susceptible to SARS-CoV-2 infection and impaired seroconversion compared to healthy controls. Follow-up data on vaccination response in dialysis patients is limited but is greatly needed to individualize and guide (booster) vaccination strategies. In this prospective, multicenter study we measured anti-spike S1 and neutralizing antibodies in 124 hemodialysis patients, 41 peritoneal dialysis patients, and 20 age- and sex-matched healthy controls over 12 weeks after homologous BNT162b2 vaccination. Compared to healthy controls, both hemodialysis and peritoneal dialysis patients had lower anti-S1 IgG antibodies (median (IQR) 7.0 (2.8–24.3) and 21.8 (5.8–103.9) versus 134.9 (23.8–283.6), respectively; *p* < 0.001 and *p* < 0.05) and a reduced SARS-CoV-2 spike protein–ACE2 binding inhibition caused by vaccine-induced antibodies (median (IQR) 56% (40–81) and 77% (52–89) versus 96% (90–98), respectively; *p* < 0.001 and *p* < 0.01) three weeks after the second vaccination. Twelve weeks after the second vaccination, the spike protein–ACE2 binding inhibition significantly decreased to a median (IQR) of 45% (31–60) in hemodialysis patients and 55% (36–78) in peritoneal dialysis patients, respectively (*p* < 0.001 and *p* < 0.05). Peritoneal dialysis patients mounted higher antibody levels compared with hemodialysis patients at all time points during the 12-week follow-up. Individual booster vaccinations in high-risk individuals without seroconversion or rapidly waning neutralizing antibody levels are required and further data on the neutralization of emerging variants of concern in these patients are urgently needed.

## 1. Introduction

The current coronavirus disease 2019 (COVID-19) pandemic poses a global threat, especially to individuals whose immune responses are diminished because of immunosuppressive therapy or diseases associated with a compromised immune system. Kidney failure and long-term hemodialysis or peritoneal dialysis treatment are associated with premature aging of the immune system and a progressive immunosenescence, resulting in both lower humoral response and T cell activity [1]. These patients are particularly susceptible to severe acute respiratory syndrome coronavirus type 2 (SARS-CoV-2) infection and more severe progression of COVID-19 compared with individuals without chronic kidney disease [2].

COVID-19 is caused by the SARS-CoV-2 virus, which consists of four major structural proteins that are called spike, envelope, membrane, and nucleocapsid protein. The spike protein includes the S1 subunit, which mediates cell surface binding via the receptor binding domain (RBD) to the host cell angiotensin converting enzyme 2-receptor (ACE2), and the S2 subunit, which induces viral–host cell membrane fusion [3,4]. Antibodies to the highly immunogenic RBD account for up to 90% of the neutralization of SARS-CoV-2-specific antibodies. However, there is little evidence of neutralizing antibodies for other SARS-CoV-2 structural proteins such as the nucleocapsid protein [3,4].

In this unprecedented time, safe and effective vaccinations were developed to counter the spread of SARS-CoV-2. One of the most commonly used vaccines is the BioNTech/Pfizer mRNA vaccine, which encodes a stabilized form of the spike protein from the SARS-CoV-2 Wuhan strain [5]. Due to their high susceptibility, dialysis patients have been prioritized for vaccination in several countries. We and others have shown that humoral and cellular immune responses are impaired following standard vaccination in hemodialysis patients [3,6]. Although most hemodialysis patients had detectable surrogate neutralizing antibodies following COVID-19 vaccination, their levels were significantly lower compared to those of healthy controls. However, because most studies focused on hemodialysis patients, knowledge about SARS-CoV-2-specific vaccine responses of peritoneal dialysis patients is still limited and determination of risk factors associated with the absence of protective neutralizing antibodies is urgently needed.

In healthy individuals, neutralizing antibody activity persists for the first few months after COVID-19 infection, but antibody levels decline over time, and vaccine-induced immunity appears to wane more rapidly than immunity induced by natural infection [7,8,9]. The emergence of new variants of concern (VOCs) poses an additional threat and an increasing challenge to our health care systems because of higher transmissibility and possible escape from vaccine-induced immunity [10,11,12]. Sustained high levels of neutralizing antibodies are a prerequisite for long-term protection against (re-)infection and severe COVID-19 [13]. Therefore, immune-compromised populations such as dialysis patients mounting lower neutralizing antibodies might become more sensitive to VOCs such as B.1.351 (beta variant) or B.1.617.2 (delta variant) [14]. Data on the longevity of vaccine-induced humoral responses in hemodialysis and peritoneal dialysis patients are urgently needed to enable individualized booster vaccinations and ultimately protect these high-risk groups permanently from severe COVID-19.

This is one of the largest prospective studies to directly compare vaccine-induced humoral responses between peritoneal dialysis and hemodialysis patients after double-dose BNT162b2 mRNA vaccination. We further determined the longevity of SARS-CoV-2 specific antibodies during a time course of 12 weeks after boost vaccination. 

## 2. Materials and Methods


**Study design and cohorts**


In this prospective, multicenter, observational cohort study, we screened 195 patients on dialysis before standard double-dose BNT162b2 mRNA (BioNTech, BNT) vaccination between December 2020 and May 2021 for eligibility at four dialysis centers in Southwest Germany. Thirty patients were excluded because they had received only a single-dose vaccination (N = 2) or had a prior COVID-19 infection (N = 28). Of the remaining 165 patients who met the inclusion criteria, 41 patients were on peritoneal dialysis, and 124 patients received hemodialysis (Figure 1). 

Anti-S1 IgG antibodies and the SARS-CoV-2 spike protein–ACE2 binding inhibition caused by vaccine-induced antibodies were determined after a median (IQR) of 22 (19–25), 20 (18–23), and 19 (19–23) days after the priming-dose and after a median (IQR) of 20 (18–21), 21 (20–24), and 23 (19–24) days after the boosting-dose in patients with peritoneal dialysis, hemodialysis, and 20 age- and sex-matched healthy controls (health care workers, N = 15, and their relatives, N = 5), respectively. Time between prime and boost vaccination was a median (IQR) of 26 (22–29), 29 (22–34), and 24 (21–27) days in patients with peritoneal dialysis, patients with hemodialysis, and healthy controls, respectively. Multiple linear regression analysis was performed to identify baseline predictors of maximum anti-S1 IgG antibody concentration in dialysis patients after the second vaccination. In addition, persistence of anti-S1 IgG antibodies and SARS-CoV-2 spike protein–ACE2 binding inhibition was measured after a median (IQR) of 91 (83–94) and 87 (85–91) days after boost vaccination in peritoneal dialysis and hemodialysis patients, respectively. 

Antibodies against the nucleocapsid protein were measured repeatedly after prime and boost vaccination and after a time course of 12 weeks after booster vaccination in all participants to exclude individuals with prior COVID-19 or breakthrough infections during follow-up. 

The study was approved by the ethics committee of the University of Heidelberg and conducted in accordance with the Declaration of Helsinki. Written informed consent was obtained from all study participants. The study is registered in the German Register for Clinical Studies (DRKS00024632). 


**Anti-SARS-CoV-2 IgG enzyme-linked immunosorbent assay**


IgG response against the spike S1 protein was determined by using the SARS-CoV-2 Total Assay (Siemens, Eschborn, Germany). The assay is a chemiluminescent immunoassay, and the defined cutoff gives a specificity of 100% and a sensitivity of 89% for detection of anti-S1 antibodies. Results are expressed as a dimensionless index, with a semi-quantitative index ≥ 1 defining positivity. 


**Detection of SARS-CoV-2 neutralizing antibodies**


The spike protein–ACE2 binding inhibition caused by vaccination-induced antibodies was determined by using a plate-based SARS-CoV-2 surrogate virus neutralizing assay (Medac, Wedel, Germany), as described previously [15,16]. The test mimics the virus–host interaction by direct protein–protein interaction using purified RBD protein from the viral spike protein and the host cell receptor ACE2. In brief, serum samples were incubated with soluble horseradish peroxidase conjugated recombinant SARS-CoV-2 RBD (HRP–RBD) to allow the binding of circulating antibodies to HRP–RBD. The mixture was then added to a capture plate, which was pre-coated with the human ACE2 protein. Unbound HRP–RBD was washed out twice, and the reactions were developed using 3,3′,5,5′-tetramethylbenzidine (TMB) as substrate. Optical density at 450 nm was measured in each well and the percent (%) inhibition of HRP–RBD:ACE2 binding was calculated as follows:$$\mathrm{Inhibition}=\left(1-\left(\frac{\mathrm{OD}\;\mathrm{value}\;\mathrm{of}\;\mathrm{Sample}}{\mathrm{OD}\;\mathrm{value}\;\mathrm{of}\;\mathrm{Negative}\;\mathrm{Control}}\right)\right)\times 100\%$$

According to the manufacturer’s instructions, a cutoff of ≥30% inhibition of receptor-binding domain:ACE-2 binding was applied. The test achieves 99.9% specificity and 95–100% sensitivity. The spike protein–ACE2 binding inhibition capacity corresponded well with the SARS-CoV-2 inhibition capacity in virus neutralization tests [15]. Therefore, vaccination-induced antibodies with spike protein–ACE2 binding inhibition are hereafter referred to as surrogate neutralizing antibodies (SNA).


**Statistics**


Data is expressed as median and IQR or number (N) and percent (%). Different groups were compared using the Mann–Whitney *U* test in the case of continuous variables and Fisher’s exact test in the case of categorial variables. When comparing more than two groups, the Kruskal–Wallis test with Dunn’s post-test was applied for continuous data and the chi-square (χ^2^) test was applied for categorial data. In dialysis patients, multiple linear regression analysis was performed to identify baseline predictors of maximum anti-S1 IgG antibody concentration 3 weeks after the second vaccination. Statistical significance was assumed at a *p*-value < 0.05. The statistical analysis was performed using GraphPad Prism version 9.0.0 (GraphPad Software, San Diego, CA, USA).

## 3. Results

### 3.1. Baseline Characteristics

From December 2020 to May 2021, we prospectively enrolled 41 patients on peritoneal dialysis, 124 patients on hemodialysis, and 20 age- and sex-matched healthy controls before a first vaccination with the mRNA vaccine BNT162b2. Baseline characteristics are given in Table 1. Antibodies against the nucleocapsid protein remained negative in all individuals during the 12-week follow-up.

Median (IQR) age at enrollment was 65 (56–78), 70 (57–80), and 60 (56–77) years for patients receiving peritoneal dialysis, patients receiving hemodialysis, and matched healthy controls, respectively (Table 1). Fifteen (37%) patients in the peritoneal dialysis group, 52 (42%) in the hemodialysis patient group, and 12 (60%) in the healthy control group were female. No statistical significance was observed for age or sex between the three different groups (Table 1). Dialysis vintage was significantly longer in patients on hemodialysis compared with patients on peritoneal dialysis with a median (IQR) of 46 (34–59) months compared to 31 (20–46) months (Table 1). 

### 3.2. Anti-S1 IgG and Surrogate Neutralizing Antibodies after First and Second BNT162b2 Vaccination in Peritoneal Dialysis Patients, Hemodialysis Patients, and Healthy Controls

After the priming dose, 22/41 (54%) of peritoneal dialysis patients, 44/124 (35%) of hemodialysis patients, and 17/20 (85%) of healthy controls developed anti-S1 IgG antibodies above the predefined threshold for positivity of ≥1. With a median (IQR) of 0.6 (0.1–1.9), the anti-S1 IgG index was significantly lower in patients receiving hemodialysis compared to both patients on peritoneal dialysis or healthy controls with a median (IQR) of 1.2 (0.5–2.7) and 5.1 (1.6–10), respectively (*p* < 0.05 and *p* < 0.001; Figure 2A). Patients receiving peritoneal dialysis had significantly lower anti-S1 IgG levels compared with healthy controls (*p* < 0.05; Figure 2A). SARS-CoV-2 SNA were detectable in 23/41 (56%) peritoneal dialysis patients, 49/124 (40%) hemodialysis patients, and 17/20 (85%) healthy controls after the priming dose. With a median (IQR) inhibition of 24% (7–46), the SARS-CoV-2 SNA were also significantly lower in hemodialysis patients compared with peritoneal dialysis patients or healthy controls with a median (IQR) of 35% (20–62) and 62% (44–75), respectively (*p* < 0.05 and *p* < 0.001; Figure 2B). No statistically significant difference was observed between peritoneal dialysis patients and healthy controls in terms of SNA after the first dose of vaccination (Figure 2B).

After the boosting dose, 39/41 (95%) peritoneal dialysis patients, 109/124 (88%) hemodialysis patients, and 20/20 (100%) healthy controls had detectable anti-S1 IgG antibodies above the cutoff. With a median (IQR) of 7.0 (2.8–24.3), the anti-S1 IgG level was again significantly lower in hemodialysis patients compared with peritoneal dialysis patients or healthy controls with a median (IQR) of 21.8 (5.8–103.9) and 134.9 (23.8–283.6), respectively (*p* < 0.01 and *p* < 0.001, respectively; Figure 2C). Peritoneal dialysis patients had significantly lower anti-S1 IgG levels compared with healthy controls (*p* < 0.05; Figure 2C). SARS-CoV-2 SNA were detectable in 38/41 (93%) peritoneal dialysis patients, 108/124 (87%) hemodialysis patients, and 20/20 (100%) healthy controls after the boosting dose. SNA were significantly lower in hemodialysis patients with a median (IQR) of 56% (40–81) compared with peritoneal dialysis patients with 77% (52–89) or healthy controls with 96% (90–98) (*p* < 0.05 and *p* < 0.001, respectively; Figure 2D). Peritoneal dialysis patients had significantly lower SNA compared with healthy controls (*p* < 0.01; Figure 2D).

After the priming dose, seropositivity for both anti-S1 IgG and SARS-CoV-2 SNA was detectable in 51% of peritoneal dialysis patients, 32% of hemodialysis patients, and 85% of healthy controls (Figure 2E). After the boosting dose, the proportion of double-positive participants increased to 92% in peritoneal dialysis patients, 86% in hemodialysis patients, and 100% in healthy controls (Figure 2E).

In addition, we performed a multiple linear regression analysis to identify baseline predictors of maximum anti-S1 IgG antibody concentration in dialysis patients 3 weeks after the second vaccination (Table 2). Higher age (β: −1.9; 95% CI: −2.6, −1.2; *p* < 0.001) was associated with a lower anti-S1 IgG antibody concentration whereas peritoneal dialysis as dialysis modality (β: 49.1; 95% CI: 29.1, 69.1; *p* < 0.001) was associated with a higher anti-S1 IgG antibody concentration (Table 2).

### 3.3. Longevity of Humoral Responses in Peritoneal Dialysis and Hemodialysis Patients after a Time Course of 12 Weeks

Twelve weeks after the second vaccination, we again measured anti-S1 IgG antibody levels and SARS-CoV-2 SNA in 41 peritoneal dialysis and 114 hemodialysis patients. Ten patients in the hemodialysis group were not available for follow-up because of death (N = 4, not COVID-19 related), hospitalization (N = 3, not COVID-19-related), and change of dialysis center (N = 3).

Anti-S1 IgG levels significantly decreased after 12 weeks compared with anti-S1 IgG levels determined 3 weeks after the boosting dose in both peritoneal dialysis and hemodialysis patients (*p* < 0.05 and *p* < 0.01, respectively; Figure 3A). Seropositivity for anti-S1 IgG antibodies decreased from 39/41 (95%) to 36/41 (88%) in peritoneal dialysis patients and from 109/124 (88%) to 88/114 (77%) in hemodialysis patients, respectively (Figure 3A). However, after 12 weeks, anti-S1 IgG levels were still significantly higher compared with levels after prime vaccination in both peritoneal dialysis and hemodialysis patients (for both *p* < 0.001; Figure 3A). 

SNA also significantly decreased after 12 weeks compared with 3 weeks after the boosting dose in both peritoneal dialysis and hemodialysis patients (*p* < 0.05 and *p* < 0.001, respectively; Figure 3B). The number of patients with detectable SNA decreased from 38/41 (93%) to 35/41 (85%) in peritoneal dialysis patients and from 108/124 (87%) to 90/114 (79%) in hemodialysis patients, respectively (Figure 3B). After 12 weeks, SNA were also significantly higher compared with levels after prime vaccination in both peritoneal dialysis and hemodialysis patients (*p* < 0.05 and *p* < 0.001, respectively; Figure 3B).

Twelve weeks after boost vaccination, hemodialysis patients had significantly lower anti-S1 IgG levels with a median (IQR) anti-S1 IgG index of 3.0 (1.1–7.8) compared with 7.0 (1.6–36.9) in patients on peritoneal dialysis (*p* < 0.05; Figure 3C). In addition, hemodialysis patients had significantly lower SARS-CoV-2 SNA compared to peritoneal dialysis patients with a median (IQR) inhibition of 45% (31–60) and 55% (36–78), respectively (*p* < 0.05; Figure 3C). 

## 4. Discussion

In this study, we demonstrate that both hemodialysis and peritoneal dialysis patients had significantly lower anti-S1 IgG and SARS-CoV-2 SNA levels after prime vaccination and after boost vaccination compared to healthy controls. However, seroconversion rates for anti-S1 IgG and SNA antibodies were high, with 86% and 92% in hemodialysis and peritoneal dialysis patients, respectively. Peritoneal dialysis patients mounted higher antibody levels compared to hemodialysis patients at all time points during follow-up. After 12 weeks, the proportion of patients with detectable SNA decreased from 87% to 79% in hemodialysis patients and from 93% to 85% in peritoneal dialysis patients, respectively.

Our data on lower seroconversion rates after SARS-CoV-2 mRNA prime and boost vaccination in hemodialysis patients compared with healthy controls are consistent with recently published studies by us and others [3,17,18]. Seroconversion rates were particularly low after mRNA prime vaccination at 15–45%, but increased to 75–95% 2–3 weeks after booster vaccination [3,17,18]. The highest SNA were observed in hemodialysis patients with a prior SARS-CoV-2 infection after only a single-dose mRNA vaccination [18]. However, although approximately 9-15% of patients with end-stage kidney disease worldwide are treated with peritoneal dialysis as first-line therapy, there are limited data on the reactogenicity and immunogenicity of available SARS-CoV-2 vaccines. Determining vaccination success is also important in these individuals, since patients on peritoneal dialysis have a reported higher mortality rate and longer hospitalization compared to the healthy general population [19]. Rodriguez-Espinosa et al. recently showed humoral response to the mRNA-1273 SARS-CoV-2 vaccine in 97% of 34 patients on peritoneal dialysis [20]. Remarkably, 63% seroconverted after only the first vaccine dose [20]. These data are confirmed by our study with detectable anti-S1 IgG and SNA in 51% after the first and in 92% after the second BNT162b2 vaccination in peritoneal dialysis patients. We show that peritoneal dialysis patients had significantly higher SNA levels than hemodialysis patients during the follow-up, but the levels were still lower compared to those of the healthy controls. Whereas higher age was associated with a lower antibody concentration, peritoneal dialysis compared to hemodialysis as dialysis modality was significantly associated with higher levels independent of the respective dialysis vintage. Of note, in contrast to our healthy control group, almost all hemodialysis and peritoneal dialysis patients suffered from comorbidities. Therefore, lower vaccine-induced antibody levels may be related, at least in part, to these comorbidities and not solely to end-stage renal disease.

To date, there are no other studies directly comparing the response to SARS-CoV-2 mRNA vaccines in hemodialysis patients, peritoneal dialysis patients, and healthy controls. However, data on immunogenicity to the hepatitis B or influenza vaccine in peritoneal dialysis patients also suggest a lower vaccine response compared to healthy controls, while cellular and humoral immunity are slightly higher compared to hemodialysis patients [21]. 

Emerging VOCs such as B.1.617.2 (delta variant) with partial immune-escape are rapidly displacing other strains worldwide and significantly higher neutralizing antibody activity is required to prevent breakthrough infections [22,23,24]. Immune-compromised individuals such as hemodialysis or peritoneal dialysis patients with lower peak antibody levels might become more sensitive to VOCs, especially with waning antibodies over time. Therefore, longitudinal data on SARS-CoV-2 vaccine-induced immune responses are essential. We showed that after 12 weeks, the proportion of hemodialysis and peritoneal dialysis patients with detectable SARS-CoV-2 specific antibodies remains high with 79% and 85%, respectively. However, SNA levels significantly decreased after 12 weeks in both groups. These values are comparable with recently published data by Stumpf et al., where 76% of dialysis patients had detectable IgG- and IgA-S1 antibodies 8 weeks after the first vaccination with either BNT162b2 or mRNA-1273 vaccines [25]. However, we recently showed that even if SARS-CoV-2 specific antibodies are detectable by commercially available tests, neutralization of VOCs may be insufficient to protect from (re-)infection, especially in individuals with low peak neutralizing antibodies [14]. Data on VOCs neutralization in high-risk cohorts with low peak and rapid waning neutralizing antibody levels are urgently needed to enable the assessment of breakthrough infection risk.

One strategy to address the decreased SARS-CoV-2 vaccine response in dialysis patients could be the administration of a third mRNA vaccine dose, as is increasingly recommended worldwide. A very recent single-center study by Ducloux et al. suggests that a third homologous BNT162b2 dose may enhance humoral responses in almost all patients, especially in those with lower antibody titers after two doses [26]. However, it remains unclear at which antibody titer and at which time point after the second vaccination a third vaccination is most promising. 

The lack of clinically validated antibody cutoffs indicating protection against COVID-19 is an important limitation of our study. However, there is increasing evidence that higher titers of neutralizing antibodies during infection are associated with lower infectivity and that breakthrough infections are more often mild or asymptomatic after initial detection of specific anti-SARS-CoV-2 antibodies [27]. Another limitation is the focus on humoral immunity without data on cellular immunity. It is also possible that individuals without detectable neutralizing antibodies are protected from infection or at least from severe COVID-19 by a cellular immune response [28]. 

Notably, the proportion of antibody-positive individuals in our study was not exactly the same for anti-S1 IgG and SNA. Tan et al. hypothesized that the discrepancy between IgG measurements and the detection of SNA might be due to the presence of neutralization synergy caused by the combination of different isotype antibodies targeting different neutralization-critical epitopes [15]. In addition, both assays were independently validated with different cutoffs to achieve the best sensitivity and specificity and may not include exactly the same patients.

In conclusion, both hemodialysis and peritoneal dialysis patients showed high seroconversion rates after the second BNT162b2 vaccination, but at significantly lower SNA levels compared with healthy controls. Of note, peritoneal dialysis patients had an enhanced humoral response compared with hemodialysis patients during the 12-week follow-up period, but SNA levels decreased significantly with time in both cohorts. As vaccine-induced antibody levels and humoral immunity decrease, individual booster vaccinations may be required in hemodialysis and peritoneal dialysis patients. Further data on neutralization of emerging VOCs are urgently needed to guide individualized vaccination strategies and ultimately protect these high-risk groups from severe COVID-19 in the long term.

## Figures and Tables

**Figure 1 vaccines-09-01130-f001:**
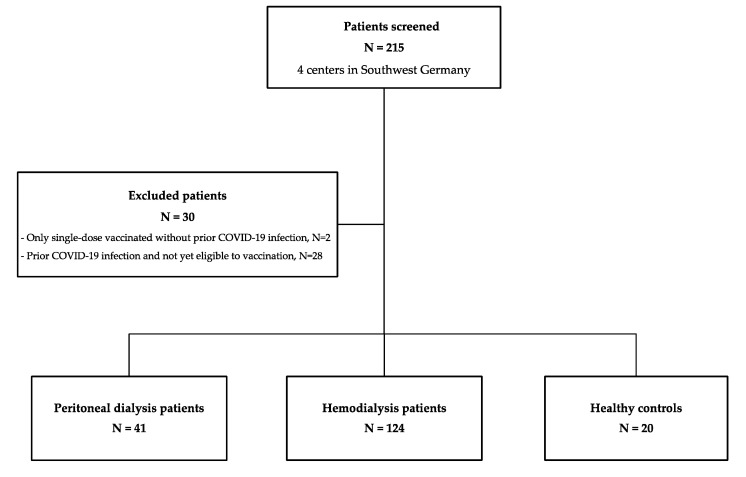
Flow chart of participating individuals. N, number.

**Figure 2 vaccines-09-01130-f002:**
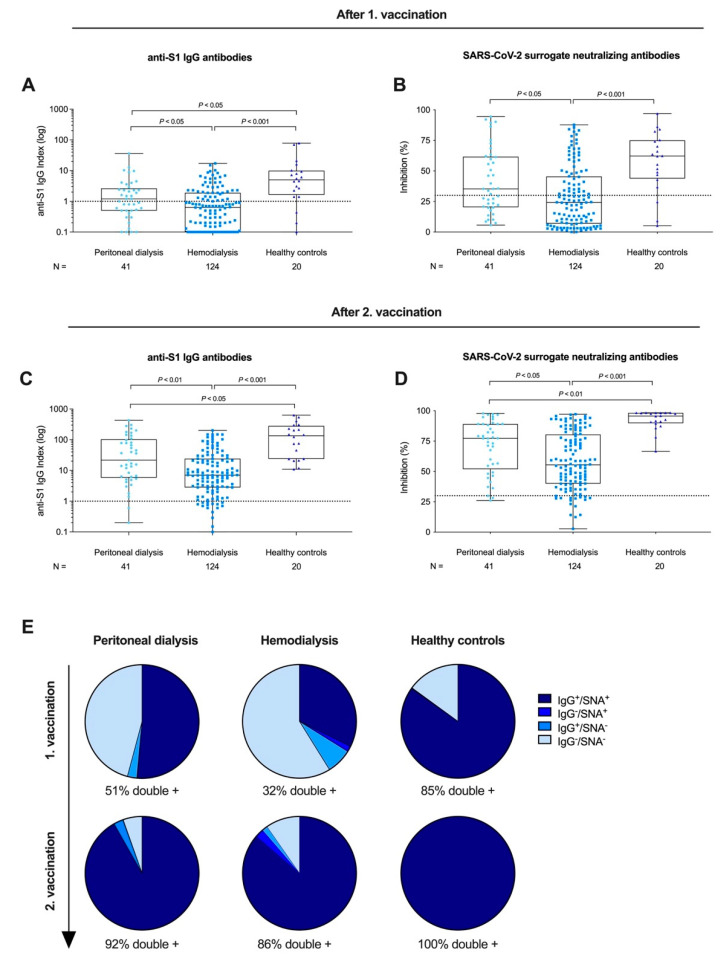
SARS-CoV-2 anti-S1 IgG and surrogate neutralizing antibodies (SNA) in patients receiving peritoneal dialysis, hemodialysis, and healthy controls after prime and boost vaccination with BNT162b2. Anti-S1 IgG antibodies after first (prime) vaccination are represented logarithmically as an anti-S1 IgG index for peritoneal dialysis patients, hemodialysis patients, and healthy controls (**A**). The dashed black line represents the cutoff for detection with a semi-quantitative index ≥1 defining positivity. SARS-CoV-2 SNA were determined by a surrogate virus neutralization test for patients receiving peritoneal dialysis, hemodialysis, and healthy controls after first vaccination (**B**). Anti-S1 IgG antibodies after second (boost) vaccination for patients on peritoneal dialysis, hemodialysis, and healthy controls. The dashed black line represents the cut-off of ≥30% in this assay according to the manufacturer’s instructions (**C**). Formation of SARS-CoV-2 SNA after second (boost) vaccination for patients on peritoneal dialysis, hemodialysis, and healthy controls (**D**). Seropositivity for anti-S1 IgG and SARS-CoV-2 SNA after first (prime) and second (boost) vaccination in patients on peritoneal dialysis, hemodialysis, and healthy controls presented as a pie chart (**E**).

**Figure 3 vaccines-09-01130-f003:**
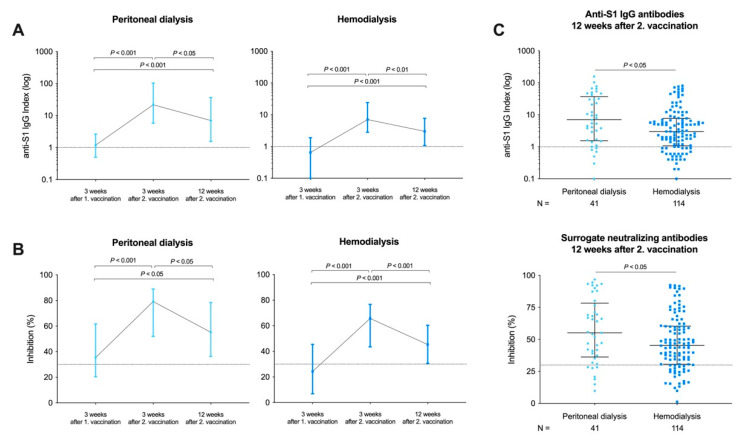
Patient-paired analysis of SARS-CoV-2 anti-S1 IgG and surrogate neutralizing antibodies (SNA) three weeks after prime, three weeks after boost, and twelve weeks after boost immunization with BNT162b2 in patients receiving peritoneal dialysis or hemodialysis treatment. Anti-S1 IgG antibodies are represented logarithmically in a patient-paired analysis for peritoneal dialysis and hemodialysis patients (**A**). Anti-S1 IgG values are given as a median with an interquartile range. The dashed black line represents the cut-off for detection with a semi-quantitative index ≥ 1 defining positivity. SARS-CoV-2 SNA is presented as percent (%) inhibition in a patient-paired analysis for peritoneal dialysis and hemodialysis patients (**B**). Anti-S1 IgG antibodies and SARS-CoV-2 SNA twelve weeks after second vaccination in peritoneal dialysis and hemodialysis patients. SNA values are given as a median with interquartile range. The dashed black line represents the cut-off of ≥30% in this assay according to the manufacturer’s instructions (**C**).

**Table 1 vaccines-09-01130-t001:** Participants’ baseline characteristics.

	Group 1 Peritoneal Dialysis	Group 2 Hemodialysis	Group 3 Healthy Controls	*p* Value
Number of patients, N	41	124	20	
Age at enrollment (years), median (IQR)	65 (56–78)	70 (57–80)	60 (56–77)	0.52
Sex (female), N (%)	15 (37)	52 (42)	12 (60)	0.21
BMI, median (IQR) Missing data, N (%)	27 (23–30) 3 (7)	25 (22–29) 11 (9)	- -	0.69
Dialysis vintage (months), median (IQR)	31 (20–46)	46 (34–59)	-	<0.001 ^a^
Cause of nephropathy Diabetes, N (%) Vascular, N (%) PKD, N (%) Glomerulonephritis, N (%) Chronic pyelonephritis, N (%) Other, N (%) Missing data, N (%)	7 (17) 10 (24) 2 (5) 9 (22) 2 (5) 9 (22) 2 (5)	29 (23) 31 (27) 3 (2) 20 (16) 2 (2) 31 (27) 8 (6)	- - - - - - -	0.40 0.94 0.43 0.40 0.24 0.69 -
Comorbidities Arterial hypertension, N (%) Diabetes, N (%) Cancer, N (%) Smoker (active/former), N (%) CAD, N (%) PAD, N (%) Chronic lung disease, N (%) Chronic liver disease, N (%) Missing data, N (%)	36 (88) 15 (37) 4 (10) 13 (32) 21 (51) 10 (24) 8 (20) 7 (17) 2 (5)	118 (95) 55 (44) 9 (7) 26 (21) 72 (58) 43 (35) 20 (16) 10 (8) 8 (6)	- - - - - - - - - -	0.10 0.38 0.61 0.16 0.44 0.22 0.62 0.10 -

BMI, body mass index; CAD, coronary artery disease; PAD, peripheral artery disease; PKD, poly-cystic kidney disease; ^a^ statistically significant.

**Table 2 vaccines-09-01130-t002:** Baseline predictors of maximum anti-S1 IgG antibody concentration in dialysis patients 3 weeks after the second vaccination.

Characteristics	B	95% CI	SE	*p* Value
Age (years)	−1.9	−2.6, −1.2	0.4	<0.001
Sex (female versus male)	10.5	−5.4, 26.4	8.0	0.20
Dialysis modality (PD versus HD)	49.1	29.1, 69.1	10.1	<0.001
BMI > 30 (yes versus no)	9.3	−10.2, 28.8	9.3	0.35
Dialysis vintage (months)	−0.1	−0.7, 0.6	0.4	0.95

CI, confidence interval of regression coefficient B; B, regression coefficient; BMI, body mass index; HD, hemodialysis; PD peritoneal dialysis; SE, standard error.

## Data Availability

The data of this study are available on request from the corresponding authors.

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
