# Peer review of "Longitudinal Humoral Responses after COVID-19 Vaccination in Peritoneal and Hemodialysis Patients over Twelve Weeks"

_vaccines, 2021, doi:10.3390/vaccines9101130_

Round 1
Reviewer 1 Report
In the current manuscript, the authors describe the formation of SARS-CoV-2 antibodies after vaccination of patients undergoing peritoneal or haemodialysis. The authors compare vaccination effect of Biontech mRNA vaccine on the formation of detectable anti-SARS-CoV-2 spike protein subunit 1 antibodies (anti-S1). Interestingly, the authors also used a test determining the capacity of patients´ serum to inhibit the binding of the spike protein with ACE2.
Major point
A flaw of the manuscript is that the authors´ from the beginning denominate antibodies mediating this inhibition as neutralizing antibodies. To me it would be appropriate to start with the description of spike protein-ACE2 binding inhibition caused by vaccination-induced antibodies. Afterwards it could be stated that the inhibition capacity of those antibodies corresponded to SARS-CoV-2 inhibition capacity in virus neutralization test (ref 12). Finally, those antibodies might be called surrogate-neutralizing antibodies (SNA or other abbreviation). Such a statement should be added to the Methods section.
In any case, a better description of the mechanism of that method is desirable. A better description might also helpful to explain why the sera of some dialysis patients inhibited spike protein-ACE2 binding although no anti-S1 IgG antibodies had been present in those sera.
Minor points
Introduction
Page 2, paragraph 2
“… VOCs such as B.1.351 (beta) or B.1.617.2 (delta) [11].”
Comment: Please use the terms “beta variant” and “delta variant”.
Materials and Methods
Page 2, paragraph 2: “Anti-S1 IgG antibodies…”
Comment: This term previously has not been introduced.
The authors should add a short paragraph to the introduction explaining that S1 is the receptor binding subunit of the spike protein and that this subunit binds to ACE2. That explanation will also facilitate understanding of the paragraph describing the detection of neutralizing antibodies on page 3.
Page 3, Figure 1
The number of individuals mentioned in the middle box and the lower boxes (peritoneal dialysis, hemodialysis and controls) is higher(N=215) than the number of individuals in the top box (N=195). Comment: Please add the control individuals to the top box.
Results
Page 4, paragraph 2
Comment: The authors state that they examined 20 healthy control with a median age of 60 years (up to 77 years old). It would be interesting to know where these individuals had been recruited, especially as neither of those had suffered from a disease (see Table 2).
Furthermore, in contrast to the healthy controls nearly all dialysis patients suffered from co-morbidities, e.g. arterial hypertension. Therefore, it is doubtful that the lower antibody response of dialysis patients solely resulted from end stage renal disease. It also seems possible that decreased antibody response was due to the co-morbidities of dialysis patients. I miss a paragraph in the Discussion dealing with this issue.
Page 7, third paragraph
“Patients with detectable neutralizing antibodies decreased from 38/41 (93%) to 35/41 (85%) in …”
Please rephrase, e.g. Number of patients with detectable neutralizing antibodies decreased from 38/41 (93%) to 35/41 (85%) in …
Page 7, last paragraph
“In addition, the SARSCoV-2-specific neutralizing antibodies were significantly lower in hemodialysis patients compared with patients on peritoneal dialysis with a median (IQR) inhibition of 45% (31–60) to 55% (36–78), respectively (P<0.05; Figure 3C).”
Comment: This sentence is hard to understand. Please rephrase.
Page 8, Figure 3 A, & B
Due to the high number of lines, the graphs in this figure are not clear. Maybe, the figure might be less confusing when showing the corresponding median values together with the interquartile range for each modality.
Discussion
Page 8, first paragraph
The statements in this paragraph already were made in the introduction. Therefore, this paragraph is unnecessary.
Page 8, first paragraph
“… B.1.617.2 (delta) rapidly displace…”
Comment: Please use the term “delta variant” when leaving that paragraph.
Page 8, first paragraph
“High levels of neutralizing antibodies after infection or vaccination are essential for long-term protection against breakthrough infections and severe COVID-19 courses.”
Comment: Please add reference(s) when leaving that paragraph.
Page 8, first paragraph
“Immune-compromised hemodialysis or peritoneal patients mounting lower neutralizing…”
Comment: Please explain the term “peritoneal patient”.
Page 8, second paragraph, last sentence
“After 12 weeks, patients with detectable…”
Comment: Please add “proportion” or “percentage”; e.g. After 12 weeks, proportion/percentage of patients with detectable…
Page 9, first paragraph
“… higher mortality and longer hospital admissions compared …”
Comment: The term admission is not appropriate. Please rephrase.
Page 9, second paragraph
“…both hemodialysis and peritoneal dialysis patients had significantly lower anti-S1 IgG antibodies and …”
Comment: Please rephrase; e.g. “…both hemodialysis and peritoneal dialysis patients had significantly lower anti-S1 IgG levels/titres antibodies and …”
Page 10, second paragraph
“Emerging VOCs such as B.1.617.2 (delta) with partial…”
Comment: Please add variant
Page 10, second paragraph
“These values are comparable with recently published data by Stumpf et al., were 76% of dialysis patients …”
Comment: Do you mean …, where 76% of dialysis patients …?
Page 10, third paragraph
“One strategy to address the decreased SARS-CoV-2 vaccine response in dialysis patients
could be the administration of a third mRNA vaccine dose, as is increasingly recommended
worldwide.”
Author Response
Reviewer 1 Comments
- A flaw of the manuscript is that the authors´ from the beginning denominate antibodies mediating this inhibition as neutralizing antibodies. To me it would be appropriate to start with the description of spike protein-ACE2 binding inhibition caused by vaccination-induced antibodies. Afterwards it could be stated that the inhibition capacity of those antibodies corresponded to SARS-CoV-2 inhibition capacity in virus neutralization test (ref 12). Finally, those antibodies might be called surrogate-neutralizing antibodies (SNA or other abbreviation). Such a statement should be added to the Methods section.
In any case, a better description of the mechanism of that method is desirable. A better description might also helpful to explain why the sera of some dialysis patients inhibited spike protein-ACE2 binding although no anti-S1 IgG antibodies had been present in those sera.
We thank the reviewer for this very important comment. We agree that although the spike protein-ACE2 inhibition capacity of vaccine-induced antibodies corresponded to SARS-CoV-2 inhibition capacity in virus neutralization tests, these antibodies should not be declared as “neutralizing antibodies”. We have taken up the suggestion of the reviewer and described these antibodies as “spike protein-ACE2 binding inhibition caused by vaccination-induced antibodies” followed by “surrogate-neutralizing antibodies (SNA)”. This was added in detail to the Methods section and also changed accordingly throughout the manuscript. The mechanisms of this method are now described in more detail.
In addition, we included a new paragraph on the discrepancy between the S1-IgG and the SNA measurements (page 9, last paragraph).
Tan et al. (ref 15) hypothesized that this discrepancy might be caused by the presence of other immunoglobulin isotypes or neutralization synergy which describes the cooperativity of different isotype antibodies targeting different neutralization-critical epitopes. This has already been observed with HIV, HCV, Ebola, and other viruses(https://doi.org/10.1128/JVI.75.24.1219812208.2001; https://doi.org/10.1016/j.celrep.2017.03.049 ; https://doi.org/10.1002/hep.27298). This may also partly explain why the sera of some dialysis patients inhibited spike-protein-ACE2 binding, although anti-S1 IgG antibodies were not present in the same sera. In addition, both assays (SARS-CoV-2 total assay, Siemens, and plate-based SARS-CoV-2 surrogate virus neutralization assay, Medac) were independently validated to achieve the best sensitivity and specificity. Thus, the respective cutoffs of >1 (Siemens) and >30% (Medac) may not include exactly the same patients which is illustrated in Figure 2E.
- Page 2, paragraph 2; “… VOCs such as B.1.351 (beta) or B.1.617.2 (delta) [11]. ”Comment: Please use the terms “beta variant” and “delta variant”.
This has been changed accordingly.
- Materials and Methods; Page 2, paragraph 2: “Anti-S1 IgG antibodies…”
Comment: This term previously has not been introduced. The authors should add a short paragraph to the introduction explaining that S1 is the receptor binding subunit of the spike protein and that this subunit binds to ACE2. That explanation will also facilitate understanding of the paragraph describing the detection of neutralizing antibodies on page 3.
We thank the Reviewer for this helpful comment. We added a basic summary of the virus’ biology with a focus on different spike proteins to the Introduction. In addition, we added reference 4 - Piccoli et al., Cell 2020.
- Page 3, Figure 1; The number of individuals mentioned in the middle box and the lower boxes (peritoneal dialysis, hemodialysis and controls) is higher(N=215) than the number of individuals in the top box (N=195). Comment: Please add the control individuals to the top box.
The control group has now been added to the top box (N=215).
- Results; Page 4, paragraph 2; The authors state that they examined 20 healthy control with a median age of 60 years (up to 77 years old). It would be interesting to know where these individuals had been recruited, especially as neither of those had suffered from a disease (see Table 2).
Furthermore, in contrast to the healthy controls nearly all dialysis patients suffered from co-morbidities, e.g. arterial hypertension. Therefore, it is doubtful that the lower antibody response of dialysis patients solely resulted from end stage renal disease. It also seems possible that decreased antibody response was due to the co-morbidities of dialysis patients. I miss a paragraph in the Discussion dealing with this issue.
Healthy controls were recruited from health care workers (N=15) and their relatives (N=5). This has been added to the Study design now (page 2, paragraph 2).
We agree that vaccine-induced antibody levels may be related, at least in part, to these comorbidities and not solely to end-stage renal disease. This paragraph/limitation has been added to the Discussion (page 9, paragraph 1).
- Page 7, third paragraph; “Patients with detectable neutralizing antibodies decreased from 38/41 (93%) to 35/41 (85%) in …”
Please rephrase, e.g. Number of patients with detectable neutralizing antibodies decreased from 38/41 (93%) to 35/41 (85%) in …
This has been rephrased as recommended.
- Page 7, last paragraph; “In addition, the SARSCoV-2-specific neutralizing antibodies were significantly lower in hemodialysis patients compared with patients on peritoneal dialysis with a median (IQR) inhibition of 45% (31–60) to 55% (36–78), respectively (P<0.05; Figure 3C).”
Comment: This sentence is hard to understand. Please rephrase.
This sentence has now been rephrased to be more precise.
- Page 8, Figure 3 A, & B
Due to the high number of lines, the graphs in this figure are not clear. Maybe, the figure might be less confusing when showing the corresponding median values together with the interquartile range for each modality.
We have modified Figures 3A and B accordingly. The values are now given as median with interquartile range.
- Discussion; Page 8, first paragraph
The statements in this paragraph already were made in the introduction. Therefore, this paragraph is unnecessary.
This paragraph has been deleted now.
- Page 8, first paragraph; “… B.1.617.2 (delta) rapidly displace…”
Comment: Please use the term “delta variant” when leaving that paragraph.
This paragraph has been deleted (see Comment 9).
- Page 8, first paragraph; “High levels of neutralizing antibodies after infection or vaccination are essential for long-term protection against breakthrough infections and severe COVID-19 courses.“
Comment: Please add reference(s) when leaving that paragraph.
This paragraph has been deleted (see Comment 9). However, this part has been cited in the Introduction now (ref 13).
- Page 8, first paragraph; “Immune-compromised hemodialysis or peritoneal patients mounting lower neutralizing…”
Comment: Please explain the term “peritoneal patient”.
This paragraph has been deleted (see Comment 9). “Peritoneal patient” was misspelled. It should have been “peritoneal dialysis patients”.
- Page 8, second paragraph, last sentence; “After 12 weeks, patients with detectable…”
Comment: Please add “proportion” or “percentage”; e.g. After 12 weeks, proportion/percentage of patients with detectable…
We added “proportion” accordingly.
- Page 9, first paragraph; “… higher mortality and longer hospital admissions compared …”
Comment: The term admission is not appropriate. Please rephrase.
The term “admission” has been replaced by “hospitalization”.
- Page 9, second paragraph; “…both hemodialysis and peritoneal dialysis patients had significantly lower anti-S1 IgG antibodies and …”
Comment: Please rephrase; e.g. “…both hemodialysis and peritoneal dialysis patients had significantly lower anti-S1 IgG levels/titres antibodies and …”
The term “levels” has been added (now on page 8, paragraph 1).
- Page 10, second paragraph; “Emerging VOCs such as B.1.617.2 (delta) with partial…”
Comment: Please add variant
The term “variant” has been added.
- Page 10, second paragraph; “These values are comparable with recently published data by Stumpf et al., were 76% of dialysis patients …”
Comment: Do you mean …, where 76% of dialysis patients …?
Thank you for this comment. We wanted to write “where”.
- Page 10, third paragraph; “One strategy to address the decreased SARS-CoV-2 vaccine response in dialysis patients could be the administration of a third mRNA vaccine dose, as is increasingly recommended worldwide.”
Unfortunately, the comment of the reviewer is missing for that point.
Reviewer 2 Report
This is a solid study comparing the kinetic of antibody levels against S1 (both by ELISA and neutralising antibodies) between healthy individuals and individuals undergoing hemo- or peritoneal dialysis over 12 weeks after vaccination with the BioNTech/Pfizer mRNA vaccine. As expected, the authors show lower antibody levels after the primary and booster vaccination as well as 12 weeks follow-up in dialysis patients compared to controls and discuss the rationale for additional booster vaccination in dialysis patients. The study contributes to a growing body of evidence demonstrating the need of additional booster vaccination against SARS-CoV-2 in a clinically vulnerable group but is essentially an extension of previous studies by the same team. Major weakness is that the antibody levels in healthy control group were not evaluated at 12 weeks post vaccination and therefore the authors cannot compare rates of decline between different groups.
Minor comments:
Authors should comment on the discrepancy between the proportion of antibody-positive individuals using ELISA and neutralisation assay after the first immunisation.
Authors should clarify that the regression analysis applies to dialysis patients only I assume.
Author Response
Reviewer 2 Comments
- Authors should comment on the discrepancy between the proportion of antibody-positive individuals using ELISA and neutralisation assays after the first immunisation.
Thank you for this comment. Please refer to Comment 1 of Reviewer 1. This has been added to the Discussion part as recommended (page 9, last paragraph).
Tan et al. (ref 15) hypothesized that this discrepancy might be caused by the presence of other immunoglobulin isotypes or neutralization synergy which describes the cooperativity of different isotype antibodies targeting different neutralization-critical epitopes. This has already been observed with HIV, HCV, Ebola, and other viruses(https://doi.org/10.1128/JVI.75.24.1219812208.2001; https://doi.org/10.1016/j.celrep.2017.03.049 ; https://doi.org/10.1002/hep.27298). This may also partly explain why the sera of some dialysis patients inhibited spike-protein-ACE2 binding, although anti-S1 IgG antibodies were not present in the same sera. In addition, both assays (SARS-CoV-2 total assay, Siemens, and plate-based SARS-CoV-2 surrogate virus neutralization assay, Medac) were independently validated to achieve the best sensitivity and specificity. Thus, the respective cutoffs of >1 (Siemens) and >30% (Medac) may not include exactly the same patients which is illustrated in Figure 2E.
- Authors should clarify that the regression analysis applies to dialysis patients only I assume.
The regression analysis applies only to dialysis patients. This has been stated appropriately throughout the manuscript now.